# Doxographical Appropriation of Nāgārjuna's *Catuṣkoṭi* in Chinese Sanlun and Tiantai Thought

**Hans Rudolf Kantor**

Graduate Institute of Asian Humanities, Huafan University, New Taipei City 223011, Taiwan;
kantorsan@hotmail.com

**Abstract:** This article reconstructs the Chinese "practice qua exegesis" which evolved out of the doxographical appropriation of the Indian Buddhist *catuṣkoṭi* (four edges), a heuristic device for conceptual analysis and a method of assorting linguistic forms to which adherents of Madhyamaka ascribed ambiguous potential. It could conceptually clarify Buddhist doctrine, but also produce deceptive speech. According to the Chinese interpreters, conceptual and linguistic forms continue to be deceptive until the mind realizes that all it holds on or distinguishes itself from is its own fabrication. Liberation from such self-induced deceptions requires awareness of the paradox that the desire to leave them behind is itself a way of clinging to them. Chinese Sanlun and Tiantai masters tried to uncover this paradox and to disclose to practitioners how the application of the *catuṣkoṭi*, on the basis of such awareness, enables proper conceptual analysis in exegesis. From this approach followed the Chinese habit of construing doxographies in which hermeneutical and soteriological intent coincide. Understanding the inner unity of doctrinal manifoldness in the translated sūtra and śāstra literature from India via exegesis also made it possible to apprehend the ineffable sense of liberation.

**Keywords:** *catuṣkoṭi*; paradox; doxography; Madhyamaka; Tiantai; Sanlun

## 1. Introduction: Deconstruction of the Conceivable and Inconceivable in Chinese Buddhism

The first indigenous Buddhist schools in East Asia—Tiantai 天台宗, Sanlun 三論宗, and Huayan 華嚴宗—emerged in the era of the Sui (581–618) and Tang (618–907) dynasties, after the transmission of Indian sūtra and śāstra literature to China had triggered a process of translation and exegetical activity. Supervised by Central Asian and Indian Buddhist masters, whose presence in China is attested since the end of the Han Dynasty (202 BC–220 AD), translation projects were usually promoted by local aristocrats or supported by the imperial court. The later schools descended from the exegetical traditions which arose due to some influential scholar monks who attempted to integrate Buddhist doctrine from India into the discourses of traditional Chinese thought.

Tiantai, Sanlun, and Huayan masters inherited from their predecessors certain methods of interpreting the translated sūtras and śāstras. The most common exegetical tool at that time was the means of doctrinal classification (doxography, taxonomy of doctrine, *panjiao* 判教). The hermeneutic purpose of this was to disclose to the Chinese practitioners the inner coherence between Indian Sarvāstivāda, Prajñāpāramitā, Madhyamaka, Yogācāra, and Tathāgatagarbha elaborations on the Buddhadharma (*fofa* 佛法, law of the Buddha). Irrespective of their differences in interpretation, all three schools resorted to the same conceptual blueprint, which was the tetralemma (*catuṣkoṭi*, *siju* 四句), frequently occurring in Kumārajīva's (344–413) Chinese translations of Indian Madhyamaka texts.

Moreover, the hermeneutical motif was inextricably linked with the soteriological intent of showing that the various textual representations of the Buddhadharma are congruent with the meaning of awakening—the mind's liberation from all self-induced deceptions. Thus, the prospected yet hidden sense of coherence was considered to be the highest

meaning of the Dharma, ultimately independent from speech and concept. The Chinese practitioners, authors, and interpreters developed an ambiguous stance toward the textual transmission of Buddhist doctrine: the independence of its ultimate meaning from speech, on the one hand, and the indispensability of the canonical word in the understanding of that meaning, on the other (Kantor 2019c, pp. 8–11).

Such ambivalence of Chinese Buddhist hermeneutics was inspired by the Indian doctrine of the "four reliances" (*catvāri-pratisaraṇāni*, *siyi* 四依) mentioned in both the *Mahāparinirvāṇa-sūtra* (T12, no. 375, p. 642, a21–24) as well as the *Da zhi du lun* 大智度論 (*Mahāprajñāpāramitopadeśa*), which Tiantai master Zhiyi 智顗 (538–597), Sanlun master Jizang 吉藏 (549–623), and Huayan master Fazang 法藏 (643–712) quoted in their own treatises and commentarial works on the sūtras and śāstras. Two of those four state: "Rely on the dharma but not on the person (*yifa buyi ren* 依法不依人); rely on the meaning but not on the words (*yiyi buyiyu* 依義不依語)" (Lamotte 1988).

Chinese interpreters derived from this their conviction that all textual manifestations of Buddhist doctrine had been designed for the multitude of non-awakened persons, while its ultimate meaning going beyond word and thought was instantiated only by the awakened—the Buddha. According to this vantage point, the canonical word and its ultimate meaning were neither identical nor separate, and the specific relationship between text and meaning implied both congruity and incongruity (D'Ambrosio et al. 2018, pp. 305–30).

For the Chinese, incongruity characterized such a relationship insofar as all linguistic expressions were considered to be false/provisional names (*Jiaming* 假名) which at best denote nominal existence but not real entities[1], while congruity was seen to be true in the specific sense that the speech of the awakened, though consisting of false/provisional names, is not misguiding, as it can unveil falsehood of which the non-awakened is not aware.[2]

Hence, two types of speech had to be distinguished: (1) the word of the Buddha (*Buddha-vacana*, *foyan* 佛言), doctrinal discourse in sūtra and śāstra (*yanjiao* 言教), which uses falsehood for heuristic purposes and thus is instructive[3], whereas (2) speech of the non-awakened unaware of its blindspots would lead listeners astray, and therefore was deemed as deceptive and called "verbal/conceptual proliferation" (*prapañca*, *xilun* 戲論, Salvini 2019, pp. 663–65; Siderits 2019, pp. 645–61). The ambivalent stance toward the transmission of the canonical word and the correspondent view about opposite types of conventional speech are concatenated characteristics of Chinese Buddhist hermeneutics, deeply rooted in the tradition of Prajñāpāramitā and Madhyamaka thought, and expressed by means of doxographies (Kantor 2019c, pp. 8–11).

For instance, the adherents of Tiantai and Sanlun believed that the inconceivable sense of liberation (*acintya-vimokṣa*, *busiyi jietuo* 不思議解脱) gives rise to the Buddha's thoughts, speeches, and physical activities, transforming the world of non-awakened beings. The inconceivable was referred to as the hidden root (*ben* 本) from which conceptual differences in terms of doctrine were believed to descend, while, conversely, the diversity of doctrine in sūtra and śāstra was regarded as the visible traces (*ji* 跡) that lead the practitioners back to the hidden root (Kantor 2019a, pp. 851–915). The Chinese constructions of doxographies aimed to outline that type of soteriological circularity. Tiantai and Sanlun interpreted liberation—the hidden meaning (*xuanyi* 玄義), ineffable ultimate, or root (*ben*)—in terms of both the source and the goal of all the differing teachings—the traces (*ji*) visible as sūtra and śāstra texts.

To enact such circularity between text and meaning, Tiantai and Sanlun emphasized the non-duality of the visible and hidden in the Buddhadharma, called the "non-duality of root and traces".[4] Reconciliation of conceptualization with the inconceivable was a major concern of the hermeneutical project in all Chinese denominations. Particularly, the dialectical approaches of Tiantai's, Sanlun's, and Huayan's doxographies focused on deconstructing the perceived duality of doctrine and liberation. The hermeneutics of

deconstructive non-duality concurs with soteriology that intends to embody the ineffable via the practice of exegesis and doctrinal discourse.

The Chinese habit of construing doxographies exemplified the specific idea of "practice qua exegesis": Buddhist wisdom, the mind's liberation from self-induced deception, had to be accomplished by specializing in exegesis of text and doctrine.[5] Wisdom-practice implicated awareness of the paradox that liberation from conceptual and linguistic hypostatization does not distance the mind from speech and thought (Kantor 2021, forthcoming).

The model which inspired the Chinese construction of doxographies was the distinction between Hīnayāna (Small Vehicle) and Mahāyāna (Great Vehicle), made by those Indian sūtras and śāstras which claimed that they themselves were promulgating the Great Vehicle. However, there was no accurate standard for such subdivision of doctrine, as it was differently defined depending on the respective denominations to which the authors or compilers of those texts were committed.

The view which was particularly influential in the Tiantai and Sanlun interpretations was that of the *Prajñāpāramitā-sūtras*, considering two major aspects of the Great Vehicle: (1) the insight that things arising and perishing in the worldly realm of sentient beings (saṃsāra) are compoundphenomena whose constituents are empty of an intrinsic nature, or, empty of inherent existence, and that this emptiness (*śūnyatā, kong* 空) is the true nature of all things which arise due to mutual dependency, called "*pratītyasamutpāda*" (*yuanqi* 緣起, conditioned arising); (2) the ultimate liberation from one's own delusions, inspiring the liberation of other sentient beings.

Chinese Tiantai, Sanlun, and Huayan doxographies assorted differing levels of Hīnayāna and Mahāyāna to interpret the variety of Buddhist concepts in a way that the supreme meaning of the Dharma can be comprehended in a gradual progress. The highest level, referred to in Tiantai and Huayan as the "round/perfect teaching" (*yuanjiao* 圓教), retained a sense of circularity: it constitutes and sublates Hīnayāna and Mahāyāna, that is, generates and suspends all doctrinal distinctions. Tiantai, Sanlun, and Huayan offered varying accounts of that thought, although it was the same conceptual figure of the *catuṣkoṭi* which had inspired them.

Western studies use "tetralemma" (four alternative assumptions) for the Sanskrit "*catuṣkoṭi*" (four edges, four points) and for its Chinese translations "four [alternative] phrases" (*siju* 四句) or "four [alternative] gates" (*simen* 四門)—briefly: "four alternatives". As for the origin, use, and philosophical meaning of the *catuṣkoṭi*, Ruegg's (1977, pp. 1–71) detailed article mentions that this scheme of classifying conceptual forms is already attested in the early philosophical literature of Indian Buddhism. The discussion in which its four alternative positions (*koṭi*) are raised concerns questions such as whether a tathāgata (Buddha) exists after death, whether the world has an end, and whether the world is eternal. Seyfort Ruegg explains: "In each of these cases the nature of a postulated entity and its relation to its predicate is investigated in such a way that all conceptually imaginable positions are exhausted; for an entity and its predicate can be conceptually related only in terms of these four limiting positions," (Ruegg 1977, p. 2).

In other words, the tetralemma (*catuṣkoṭi, siju*) is a formal scheme of four mutually related yet distinct ways of referencing a specific doctrinal topic; it often (but not always) was considered to provide an exhaustive set of four mutually exclusive conceptual possibilities (Priest 2018, p. 23; Westerhoff 2005, pp. 367–95): (1) the application of a certain concept (2) the application of its negation (3) the application of both the concept and its negation, and (4) the application of neither the concept nor its negation.

In Indian Prajñāpāramitā and Madhyamaka literature, particularly in Nāgārjuna's *Mūlamadhyamaka-kārikā* (*Zhong lun* 中論), there is both (a) the positive (cataphatic) and (b) the negative (apophatic) application of the four alternatives: (a) all the four are asserted; (b) all of them are rejected (Westerhoff 2005, pp. 367–95). In the Chinese interpretation of Sanlun master Jizang, the apophatic and cataphatic versions of the *catuṣkoṭi* indicate a strong tendency to merge into one another. In his commentary on Nāgārjuna's *Kārikā*—the *Zhongguan lun shu* 中觀論疏, Jizang's dialectical notion of "suspension (sublation) of the

four alternatives" (*jue siju* 絶四句) highlights the thought that it is the same principle which constitutes and deconstructs all four.

Moreover, Tiantai, Sanlun, and Huayan developed interpretations in which all positions of the cataphatic *catuṣkoṭi* appear to be equally valid. Zhang's recent article explains that the *catuṣkoṭi* can be described as a Buddhist form of dialectics which "consists of four arguments or perspectives, only one of which can be true at a time" (Zhang 2016, pp. 25–71). According to the Chinese interpretations, however, this cannot be mistaken in the way that only one perspective apart from the other three is valid, as none of the four represents an independent logical parameter. The position that excludes the other three is a construction leading the mind astray. Sanlun, Tiantai, and Huayan texts emphasize that such misconception is a symptom of the clinging and non-awakened mind.[6]

Therefore, Chinese Buddhist traditions regarded the linguistic–conceptual figure of the *catuṣkoṭi* as a symbol for both the awakened understanding of the textual transmission of Buddhist doctrine and its exact opposite—the non-awakened mind inconsonant with, or unaware of the Buddhadharma. The non-awakened mind was considered to be impeded by or entangled in its own conceptual constructions.

In his classic on mind-contemplation (*guanxin* 觀心), the *Mohe zhiguan* 摩訶止觀, Tiantai master Zhiyi, like Sanlun master Jizang in his commentary on the *Vimalakīrti-nirdeśa-sūtra*, the *Jingming xuanlun* 淨名玄論, developed the list of "ten types of *catuṣkoṭi*" (*shizhong siju* 十種四句) to demonstrate that our understanding of the four alternatives must not be confined to only one valid reading. The proper understanding must learn to see that it is the same *catuṣkoṭi* which can be either (1) the source of misguiding conceptualization or (2) the heuristic and analytic device for those who seek liberation from such deception.

Hence, the *catuṣkoṭi* in Tiantai, Sanlun, and also Huayan figures as a linguistic symbol which mirrors the ambivalence of conventional speech that could be divided, on the one side, into doctrinal discourse (*yanjiao*) as transmitted in sūtra and śāstra and, on the other, into verbal or conceptual proliferation (*prapañca*, *xilun*) of those who are unaware of the Buddhadharma. Those opposites were together epitomized through the same symbol of the *catuṣkoṭi*.[7]

Seen from this vantage point, the ambiguity of the *catuṣkoṭi* can be compared to the notion of "pharmaca". In spite of their therapeutic purpose, pharmaca can be toxic. Their functioning as medicine does not eradicate their potential for poison, just as their poisonous effect retains potential for being medicine. Each has the potential to be its own reverse. The two are opposites yet inseparably one. The *catuṣkoṭi* is a symbol for such ambivalence in conventional speech. For the Chinese interpreters, the non-duality of the salutary and harmful, symbolized by the *catuṣkoṭi*, provides the condition that allows the deluded mind to transform itself into the opposite, liberation.

On account of such ambiguity, Buddhist masters in China were able to develop their respective doxographical frameworks (*panjiao*). In what follows, the article examines (1) Nāgārjuna's *catuṣkoṭi*, as it figures in the Chinese transmission of Madhyamaka, and (2) the specific influence of this analytic method on the formation of doxography and the associated practice of hermeneutical deconstruction in Chinese Tiantai and Sanlun exegesis.

## 2. Nāgārjuna's Application of the Catuṣkoṭi According to Kumārajīva's Transmission

Nāgārjuna's main work, the *Mūlamadhyamaka-kārikā* (*Root Verses on the Middle Way*), was translated by Kumārajīva in 409. Its Chinese title is called *Zhong lun* 中論 (*Treatise on the Middle*), and besides Nāgārjuna's verses, the translation also contains one of the oldest existent commentaries ascribed to a person called Qingmu 青目 (*Vimalākṣa) or Binjialuo 賓伽羅 (*Piṅgala). The compiled text of the *Zhong lun*—Nāgārjuna's verses and Piṅgala's commentary, neither existent in Sanskrit nor in Tibetan—was highly influential throughout the history of Chinese Buddhist thoughts. Jizang's and Zhiyi's works borrowed a lot from this early Indian Madhyamaka treatise; particularly, the use of the *catuṣkoṭi* was relevant to the development of Tiantai and Sanlun. The Chinese masters usually did not differentiate between Nāgārjuna and Piṅgala when they quoted from Kumārajīva's *Zhong lun*.[8]



All Chinese schools took Nāgārjuna's positive (cataphatic) and negative (apophatic) applications of the four alternatives in account and construed their doxographies on that basis. As for the apophatic or negative sense, the first chapter of Nāgārjuna's *Mūlamadhyamaka-kārikā* (*Zhong lun*) provides an example which has been very influential in Tiantai and San-lun thoughts. In that chapter, Nāgārjuna investigates the notion of "condition" (*pratyaya*, *yuan* 緣), defined as a certain thing which has the capacity to give rise to another thing. However, in his discussion, he rejects the four possibilities of "arising" (*utpāda*, *sheng* 生): arising on account of either (1) itself, or (2) something else, or (3) both, or (4) neither (*Zhonglun*, T30, no. 1564, p. 2, b6–17).

Assuming that, in this way, all conceptual possibilities are exhausted, he concludes that—even in reference to "*pratītyasamutpāda*" (*yuanqi*, conditioned arising)—no real arising can ultimately be asserted. There is no real entity which arises and perishes. In a paradoxical fashion, he describes the nature of conditioned arising as "non-arising" (*anutpāda*, *bushing* 不生); another term for this is "emptiness" (*śūnyatā*, *kong* 空). The commentator Piṅgala explains that such sense of negation—the exclusion of all four conceptual possibilities—equals the "ultimate meaning" (*diyi yi* 第一義) which is emptiness of inherent existence, the ultimate sense of true emptiness (*Zhong lun*, T30, no. 1564, p. 1, c12).[9]

However, what is said about "arising" also applies in reverse, or is true of the opposite "non-arising," that is, "emptiness". Hence, another well-known example of the tetralemma's negative application exactly expresses this in chapter 22: (1) emptiness (non-arising) should not be asserted, (2) nor non-emptiness, (3) nor both, (4) nor neither (*Zhonglun*, T30, no. 1564, p. 30, b22–23). Nāgārjuna explains: "It is only said for the purpose of provisional designation (*dan yi jiaming shu* 但以假名説)," (*Zhonglun*, T30, no. 1564, p. 30, b23). These two examples of denying all four alternatives are exclusive negations of an exhaustive set of conceptual possibilities concerning a topic that has been considered as crucial throughout all doctrinal discourse in the history of Buddhism, namely, "arising" and the opposite "non-arising" (emptiness).[10]

Nonetheless, as Nāgārjuna's last remark stresses, the four positions might alternately and provisionally be employed in changing contexts of doctrinal discourse. What legitimizes the provisional use of any of these four in conventional speech is the rejection of misunderstanding all in the ultimate sense. No form of linguistic signification applies ultimately. The negative application of the tetralemma is a linguistic device for the exclusion of ultimate meaning in the domain of speech. Negation in the sense of what asserts the opposite is to be negated, too.

Hence, Westerhoff (2005, pp. 367–95, 2009, pp. 69–92) and also Zhang (2016, pp. 25–71) correctly observe that such excluding sense is not the type of negation used to construe correlative opposites such as "emptiness and non-emptiness," or "non-arising and arising". Nāgārjuna's "non-arising" in the first chapter excludes not only arising but also non-arising; in his commentary to Nāgārjuna's *Kārikā*, the *Zhongguan lun shu* 中觀論疏, Jizang expands on this thought (T42, no. 1824, p. 11, a28–b20; p. 28, c8–19).[11]

However, Westerhoff's observation is inspired by Ruegg's (1977, pp. 43–47) hint at Candrakīrti's commentary to the *Kārikā*, the *Prasannapadā*, in which the sixth-century Indian commentator says that the exclusive sense of negation represents the case that traditional Sanskrit grammarians call "*prasajya-pratiṣedha*" (non-affirmative negation); in later Chinese translations it appears as "*zheqian* 遮遣". Westerhoff suggests for the Sanskrit term "exclusion negation".

Different from "*prasajya*", negation used to construe and assert correlative opposites, or negation which, for instance, implicitly validates the opposite of what is negated, such as non-being in contradistinction to being, or non-arising contrasted with arising, might correspond to the grammatical term "*paryudāsa-pratiṣedha*" (implicative negation), in Chinese known as "*zhebi* 遮蔽". Westerhoff understands this differently, translates "*paryudāsa*" as "presupposition negation," and calls "*prasajya*" sometimes also "cancelling presuppositions".[12]

Although Nāgārjuna's *Kārikā* does not use either of the grammatical terms (*prasajya* or *paryudāsa*), Westerhoff's attempt to understand the negative application of the *catuṣkoṭi* in terms of this distinction seems to be helpful. The external negation of all four *koṭis* can be considered as *prasajya* negation, differing from the internal negations of the second and fourth alternatives within the *catuṣkoṭi* itself. Those two (the second and the fourth) could, perhaps, be subsumed under *paryudāsa* negation. Following this distinction between internal and external negation, or perhaps *prasajya* and *paryudāsa*, we might say that the first and the second of the four *koṭis* (just like the third and the fourth), relate to one another as correlative opposites.[13]

This is to say, the negation in the second *koṭi* asserts the correlative opposite of the first, just as the first is the correlative opposite of the second and therefore could be regarded as the negation of the second: emptiness in the first *koṭi* is the negation of non-emptiness in the second and vice versa; self-arising is the opposite of arising from something else (not from itself) and vice versa. Hence, negation in the second *koṭi* (denying the first) seems to function as a *paryudāsa* negation referencing what pertains to the domain of speech and thought.

This is different from *prasajya* negation which rejects the four alternatives altogether and does not allow for any affirmative form of linguistic signification. The same sense of *paryudāsa* negation seems to apply to the relationship of the third and fourth *koṭi*, as the fourth is meant to deny the third and vice versa (see Piṅgala's and Bhāviveka's discussion in Section 3). What seems to constitute each of these four as distinct elements pertaining to the same set is the relationship of mutually dependent or correlative opposites (*xiangdai* 相待).[14]

As to the conceptual structure of the two mentioned instances of the *catuṣkoṭi* in the *Zhong lun*, *paryudāsa* negation could be considered a constitutive factor for such types of interdependency, that is, the two opposites (the first and the second *koṭi*) negate each other, and in such mutual negation they mutually assert. To deny one side is to assert the other, which means without its opposite neither side can be held. In denying its opposite, each of the two relies on the other.

This precisely is what generates the third *koṭi*. As neither the first nor the second can be held apart from the other, the third comes to the fore, embracing the entire relation of those two. However, again, it is the same principle of constituting correlative opposites that also applies to this third case. What one side asserts is the opposite of the other—to assert one side is to be the opposite of its own negation, which yields the fourth as that which is the implicit opposite of the third.

Consequently, the fourth excludes what the third includes; the fourth is neither the first nor the second. This could continue infinitely, as has been pointed out by the Chinese Tiantai and Sanlun interpretations as well as by the *Da zhi du lun* (T25, no. 1509, p. 259, c29-p. 260, a2), which even talks about five alternatives (*wuju* 五句)—the abandoning of all four (*she shi siju fa* 捨是四句法) is the fifth. However, again, this would trigger the abandoning of abandoning, etc., and thus elicit the endless repetition of what has been displayed by the preceding positions, that is, what has already become evident with the fourth *koṭi*. In such a sense, the *catuṣkoṭi* may represent an exhaustive set of four conceptual possibilities which are mutually exclusive (or distinct).

For Nāgārjuna, all this means that, although there is no real entity to which we could attribute the property of arising, "non-arising" or "emptiness" (only denying a real entity which arises) must not exclude the meaning of unreal arising. This is so because unreality (emptiness of inherent existence) is not equivalent to nonexistence or nothingness. What is unreal might either be deceptive or instructive and therefore is existentially relevant, and yet it is not what can really be appropriated or dismissed. Hence, in a limited, provisional sense, or salutary context, unreal arising functions as a heuristic means that should not and cannot be denied.

Ultimately, the true sense of non-arising, the emptiness of inherent existence, means to be empty of an irreducible or real core; this even applies to emptiness. However, linguistic

signification, which tends to confuse its referents with independently existent entities, may evoke an image of reality, falsely attributed to "emptiness," if "emptiness of emptiness" (*kongkong* 空空) is not seen in conditioned arising. Because of this paradoxical sense of "emptying emptiness again" (*kong yifu kong* 空亦復空) there is no fixed linguistic form which can truly predicate the ultimate sense (*Zhong lun* T30, no. 1564, p. 33, a20–21). The true meaning must be comprehended by paradoxically denying what this expression refers to.

The paradoxical differentiation of ultimate truth from conventional truth, expounded in chapter 24, embodies this sense and must be applied to all doctrinal conceptualization.[15] Objects of our epistemic–propositional references might be considered as conventionally true in the sense that they are things that arise due to causes and conditions, and yet all this is empty of inherent existence and unreal. Confined to conventional truth, such (unreal) "arising" evades ontological determinacy in terms of either being, or non-being, or both, or neither. At the level of speech and thought, the distinction between conventional and ultimate truth is therefore necessary, and yet such differentiation does not really reach beyond the linguistic level. The distinction is self-inclusive, sublating itself due to the emptiness of what is differentiated. Differentiation entails non-duality.[16]

Linguistic expression, unaware of this paradox, is prone to confuse unreality with reality. Madhyamaka's paradox of non-duality qua distinction could be rephrased asthe following: To separate the conventional from the ultimate means to mistake the unreal for the real, whereas not differentiating between the two undermines the true sense of emptiness in conditioned arising. In order to see non-duality, incessant distinction is necessary. That the two truths are neither identical nor separate is a paradox but not a contradiction, because contradiction would mean the opposite: identity and separation of the two at the same time, which contradicts the true sense of emptiness.[17]

Nāgārjuna's negation of both real arising and real non-arising implies the paradoxical distinction of the two truths. The two negative forms of the *catuṣkoṭi* are applied to validate a certain type of conventional speech—one which advances awareness of the unreality of its own referents. Critical validation, rooted in invalidation, shows the limits of thought and speech, thereby revealing the heuristic value of language and analysis. This is believed to be the type of speech—"the supreme among all types of speech" (*vadatām varam*, *zhushuozhong diyi* 諸中説第一) leading to the pacification of conceptual proliferation (*prapañcopaśama*, *mie xilun* 滅戲論). The first verse in the *Kārikā* (*Zhong lun*) literally expresses this (Siderits and Katsura 2013, p.13; *Zhong lun* T30, no. 1564, p. 1, c11).

On that basis, the positive application of the tetralemma can fulfill the same purpose that is ascribed to the negative application. The cataphatic use of the third instance of the *catuṣkoṭi* in chapter 18 explicates this: "All is real, unreal, both real and unreal, neither real nor unreal. This is called dharma of all Buddhas. 一切實非實, 亦實亦非實, 非實非非實, 是名諸佛法". (*Zhong lun* T30, no. 1564, p. 24, a5–7).[18]

If we look at the pre-modern commentarial literature, authored by Piṅgala, Bhāviveka, Candrakīrti, as well as the Chinese Tiantai and Sanlun interpreters, Zhiyi and Jizang, the positive version of this specific *catuṣkoṭi* has been interpreted in various ways, all of which can nevertheless be subsumed under three associated aspects: (1) the differentiation of the two truths with respect to the first and second *koṭis* which appear as opposites, (2) the relationship between the two in terms of correlative dependency, and (3) the gradation of the Buddha's teaching, which, in the case of the Chinese interpreters, led to their classifications of Buddhist doctrine (doxography).

## 3. Piṅgala's and Bhāviveka's Interpretations of Nāgārjuna's Catuṣkoṭi in the Chinese Sources

Although Piṅgala (4th AD) and Bhāviveka (500–560) equally hold that the first two *koṭis* in Nāgārjuna's positive application account for the differentiation of the two truths, they expound their views in opposite ways. For Piṅgala, the first *koṭi* represents ultimate truth (ultimate meaning, *diyi yi* 第一義), because everything is real (*yiqie shi* 一切實) in

the specific sense that it is ultimate and true emptiness which is the real nature (*shixing* 實性) wherein all distinct and inexhaustible forms of conditioned arising are established; he explains:

> Question: If the Buddha does not talk about the self and non-self, causing all fabrications of the mind to extinguish and bringing all speech to an end, how can he make persons become aware of the real mark of all dharmas? Answer: All Buddhas resort to the skill in devising infinite expedient means, and all dharmas are [ultimately] empty of a determinate mark. On that basis, and for the sake of saving the multitude of sentient beings, they sometimes teach that everything is real, or at another time explain that everything is unreal, or at a different moment say that everything is both real and unreal, or at some other occasion hold that everything is neither real nor unreal. [If they say that] everything is real, they set forth the real nature of all dharmas, so that all realize the ultimate meaning and the single mark of equality, called emptiness of any mark. This is like all the streams with varying colors and flavors [of water] flowing into the great ocean in which they become the single color and the single flavor. 問曰: 若佛不説我, 非我, 諸心行滅, 言語道斷者, 云何令人知諸法實相? 答曰: 諸佛無量方便力, 諸法無決定相, 為度眾生或説一切實, 或説一切不實, 或説一切實不實, 或説一切非實非不實. 一切實者, 推求諸法實性, 皆入第一義平等一相, 所謂無相, 如諸流異色異味入於大海則一色一味. (*Zhong lun* T30, no. 1564, p. 25, a14–20).

What the first *koṭi* predicates of ultimate truth must be seen in the context of conventional speech. The manner in which it represents ultimate truth depends on the other three; the same mutual dependency of all four applies to each (see the succeeding part of this quote below). This is so, because the emptiness of an intrinsic nature is the real nature of all, which is to be empty of a determinate mark—in this sense, "everything is real".

To mistake this first predication for the fixed form of the ultimate, excluding the other positions, contradicts its very meaning. Empty of a determinate mark, the ultimate is to be predicated in a way which reveals that there is no fixed predication of it. As this is true of all, the text illustrates the real nature with the single flavor, into which all differing flavors turn.

In such sense of mutual dependency, the second *koṭi* accounts for the opposite of the first, which is the perspective of the worldly realm before having attained the highest sense of the ultimate. This is unreality (*feishi* 非實) in the shape of the differentiated and multifaceted conventional truths that might lead to the ultimate in the same way as all flavors of water in the differing streams are capable of becoming the single flavor of the ocean. Using a similar image, the *Da zhi du lun* explains the relationship between the conventional and the ultimate in exactly this fashion.[19] Piṅgala goes on:

> [If the Buddhas say that] everything is unreal, [they mean that], before all dharmas dissolve into the real mark, each distinct view [of them] is empty of any real core, as all there is, only exists due to the manifold conditions which come together. [If they say that] everything is both real and unreal, [they observe] that the multitude of sentient beings can be divided into three types: those with the higher, middle, and lower [faculties]. Those with the higher [faculties] see that the marks of all dharmas are neither real nor unreal. Those with the middle [faculties] see that the marks of all dharmas are both real and unreal. Because of the shallowness of their wisdom, those with the lowest [faculties] see that the marks of all dharmas contain a small proportion of what is real and also a small proportion of what is unreal. When they observe that nirvāṇa is an unconditioned dharma and indestructible, they perceive what is real; when they observe that saṃsāra is a conditioned dharma and deceptive, they perceive what is unreal. [When the Buddhas say that] everything is neither real nor unreal, they do so because they deconstruct [such dual views about] what is real and what is unreal. Question: At another occasion the Buddha has said that one should

distance oneself from [the view of] neither being nor non-being, why is it now said that what the Buddha has taught is neither being nor non-being [neither real nor unreal]? Answer: It was a different context when [the Buddha taught] the deconstruction of all the four types of clinging, [the apophatic *catuṣkoṭi*], but now in [Nāgārjuna's exposition of] the four alternatives, [the cataphatic *catuṣkoṭi*], there is no entanglement in deceptive speech and thought. Once he heard the Buddha's teaching, he attained the path of awakening, therefore he said "neither real nor unreal". 一切不實者, 諸法未入實相時, 各各分別觀皆無有實, 但眾緣合故有. 一切實不實者, 眾生有三品, 有上, 中, 下. 上者觀諸法相非實非不實. 中者觀諸法相一切實一切不實. 下者智力淺故, 觀諸法相少實少不實; 觀涅槃無為法不壞故實; 觀生死有為法虛偽故不實. 非實非不實者, 為破實不實故, 説非實非不實. 問曰: 佛於餘處, 説離非有非無; 此中何以言非有非無是佛所説? 答曰: 餘處為破四種貪著故説, 而此中於四句無戲論. 聞佛説則得道, 是故言非實非不實. (*Zhong lun* T30, no. 1564, p. 25, a20–b2).

The passage as a whole illustrates that clinging to either of the four, while dismissing any of the other three, is what would lead practitioners of Buddhist doctrine astray. Therefore, Piṅgala explains that the Buddha, at another occasion, applied the apophatic *catuṣkoṭi*, denying also the fourth position, and yet what the Buddhameant is the same that Nāgārjuna presents in his cataphatic exposition. If the third *koṭi* were misunderstood so that the other positions were not taken in account, it would misrepresent the two truths as duality, contravening the meaning of true emptiness. However, integrated into the entire *catuṣkoṭi*, the third might correctly point at the necessity to differentiate between two truths, that is, between saṃsāra and nirvāṇa.

Furthermore, Piṅgala clarifies that, in order to undermine any mistaken sense of duality, the fourth only fulfills the purpose to deconstruct the third. Although opposed to the third, the fourth is not meant to completely separate from the preceding three. If this view arises in the practitioners' minds, they are advised to distance themselves from the fourth. What helps in this case is the "the deconstruction of all four types of clinging"—Nāgārjuna's apophatic application of the *catuṣkoṭi*.

Overall, Piṅgala's commentary shows that the apophatic and cataphatic versions of the *catuṣkoṭi* are not in contradiction with one another; the two represent the same. Yet, their relationship is paradoxical: although the *catuṣkoṭi* may heuristically function as a means of doctrinal analysis, it simultaneously bears the risk of becoming the source of entanglement in deceptive speech and thought. It undeniably is a medicine, but one whose toxic potential must not be ignored; to reveal its ineradicably harmful potential is to bring about its therapeutic effect.

In the Chinese *Boredeng lun shi* (*\*Prajñāpradīpa*) translated by Prabhākaramitra (565–633), Bhāviveka appears to choose the opposite approach: the first position in the same *catuṣkoṭi* seems to represent a view reminding us of Sarvāstivāda: the eighteen *dhātus* (*shiba jie* 十八界) are the irreducible elements which constitute all compound phenomena that arise in the realm of saṃsāra.[20] In contrast to those unreal phenomena, these elements are considered as real. However, for the Mādhyamikas, such a concept of saṃsāra is incomplete, as it lacks a thorough understanding of emptiness. Thus, it does not reach beyond the realm of conventional truth; "everything is real" (*yiqie shi* 一切實) in the first *koṭi* implies confinement to the domain of the conventional truth, in contrast to what the *Zhong lun* represents as Piṅgala's view of the ultimate truth.

Developing a deeper understanding, the second *koṭi* points at emptiness, revealing unreality (*feishi* 非實) in all conditioned arising. This also applies to the eighteen elements. Emptiness is ultimately true in the sense that all arising is unreal. Unreality is seen from the higher vantage point of ultimate truth. Although Piṅgala and Bhāviveka interpret the first and the second *koṭi* in a reverse manner, they agree that the two positions represent the two truths as opposites. Bhāviveka's interpretation seems to introduce a view of progression: first, the lower conventional and, second, the higher ultimate. The third *koṭi* reveals that the two truths are correlatively dependent. This is referred to as "both real and unreal".



The last step is the fourth *koṭi*, adumbrating full awakening as a state of mind devoid of clinging and conceptual distinction; this is called "neither real nor unreal".[21]

Overall, Bhāviveka's interpretation of the *catuṣkoṭi* seems to disclose to practitioners the ascending and graded levels of the Buddha's teaching. He proceeds with this project by gradually deconstructing the stratified forms of clinging in their conceptual attitudes until full awareness of all fabrications in thought and speech is accomplished. Perhaps, it is the distinction between Mahāyāna and Hīnayāna which has inspired his view: the first *koṭi* may represent Sarvāstivāda as the lower understanding of Hīnayāna, while the succeeding levels describe the ascension to the higher stages of Mahāyāna until awakening is completed, which culminates in transcending all (deceptive) distinctions. Candrakīrti seems also to stress a similar reading of doctrinal gradation, yet without mentioning the two truths (Ruegg 1977, p. 23; Westerhoff 2005, p. 34).

The Sanskrit version of Bhāviveka's *Prajñāpradīpa* is not extant; only the Chinese and Tibetan translations have been transmitted. Jizang and Zhiyi were not aware of Bhāviveka's thought, and later Tiantai tradition did not pay attention to the translations from the Tang dynasty. By contrast, Piṅgala's commentary exerted significant influence on both Zhiyi and Jizang, although the two Chinese interpreters critically remarked that his annotations are one-sided. Interestingly enough, Bhāviveka's outline of the gradated structure of Buddhist doctrine yet demonstrates a remarkable degree of conceptual affinity with the Chinese patterns of doxography. Besides Bhāviveka's taxonomic approach to the *catuṣkoṭi*, his discussion also offers another reading from a soteriological point view, similar to Piṅgala's commentary, which shows that, for him, the concept of *catuṣkoṭi* must be treated with a sense of ambiguity (*Boredeng lun shi* T30, no. 1566, p. 108, a4–b2).

## 4. Chinese Sanlun and Tiantai Interpretations of Nāgārjuna's Catuṣkoṭi

In the Chinese transmission of Indian sūtra and śāstra literature, this third instance of the *catuṣkoṭi* appears not only in Kumārajīva's translation of Nāgārjuna's *Kārikā*, the *Zhong lun*, and the *Boredeng lun shi*, but also in the the *Da zhi du lun* (not existent in Sanskrit or Tibetan), which expands on it in tandem with the two truths and another associated concept, called the "four *siddhāntas*" (*si xitan* 四悉壇) (T25, no. 1509, p. 61, b6–18).

The Sanskrit term "*siddhānta*" means established doctrine, and the Chinese created a transliteration for this. The four *siddhāntas* are an extension of the two truths. In terms of doctrinal debate, "four *siddhāntas*" only played a significant role in Tiantai tradition. The last of the four accounts for the supreme or ultimate meaning (*diyi yi* 第一義) which instantiates "*shixiang* 實相", rendered as real mark, that is, the nature of reality. One of the Sanskrit terms for which ancient translators construed those Chinese expressions was "*dharmatā*"—literally: dharma nature (nature of all things). Chinese Buddhist texts used for dharma nature also the literal translation "*faxing* 法性".

In Tiantai and Sanlun, "*shixiang*", "*faxing*", and "*diyi yi*" were often treated as synonyms. According to the Chinese transmission and interpretation of Prajñāpāramitā and Madhyamaka, the nature of reality equals the mind's liberation from its self-induced deceptions, as the undistorted mind sees all things that it encounters in the way they actually are. This also includes its own nature which many Tiantai masters interpreted as the nature of reality (real mark, dharma nature, ultimate meaning)—if what is seen by the unobstructed mind is identical to what things truly and really are, "true and real" (*zhenshi* 真實) can be predicated on both what is seen and what is seeing, that is, such differentiation must dissolve in the real mark. This is ultimate meaning.

Like other Prajñāpāramitā and Madhyamaka texts, the *Da zhi du lun* stresses that ultimate meaning or real mark (*shixiang*) evades linguistic representation. However, mentioning the two truths in association with the *catuṣkoṭi*, the text seems not to imply that ultimate meaning or real mark must completely separate from speech and thought. Instead, the proper way of approaching it is to neither separate from all four, nor to cling to any apart from the others. To consider all four together is to enter the state of mind which realizes that all the objects it clings to or separates from are its own fabrications—the

liberated mind sees that separating entails clinging! In this paradoxical sense, ultimate meaning or real mark instantiated by the liberation of the mind from deception does not separate from speech and thought, although there must be no clinging to a fixed form of linguistic expression; (*Da zhi du lun* T 25, no. 1509, p. 61, b6–18).

Jizang's Sanlun works and Zhiyi's Tiantai texts quote this *catuṣkoṭi* and comment on it several times, using the version from the *Da zhi du lun*, which they regard as identical to that of the *Zhong lun* (*Kārikā*). In his commentary on the *Kārikā*, the aforementioned *Zhongguan lun shu*, Jizang observes ontological indeterminacy when examining the question of what the nature of reality (*shixiang*—real mark) means. In his discussion, he employs a binary which frequently occurs in Chinese exegesis of sūtra and śāstra. This binary, most probably resulting from a fusion of indigenous thought and Buddhist concepts, started to appear in the sixth century and, in the present context, might best be translated as "cohesive body and functions" (*tiyong* 體用).[22] According to Jizang, the Chinese term "real mark" ("*shixiang*" probably used for the Sanskrit "*dharmatā*") must be understood through these two aspects, that is, the ultimate meaning of Buddhadharma, the sense of liberated mind, becomes accessible to practitioners by perceiving the cohesive body in the functioning of all the four alternatives. He explains:

> These four gates [*catuṣkoṭi*] are the expedient means [*kauśala-upāya*] bound for the real mark [*dharmatā*]. To let one's mind, [in an unimpeded manner], wander through all four gates is to enter the real mark [*dharmatā*]. Hence, the four gates display the functions [of the real mark], and what is not divided into four [within all four] is the cohesive body [of the real mark]. 此之四門, 皆是實相方便. 遊心四門便入實相. 故以四門為用, 不四為體; (*Zhongguan lun shu* T42, no. 1824, p. 124, a12–23).

Awareness of ontological indeterminacy involves the previously mentioned paradox of distinguishing two truths (non-duality qua/yet differentiation); "cohesive body and functions" is analogous to "ultimate truth and conventional truth". In a preceding passage, Jizang explains that the *catuṣkoṭi* (four gates, *simen*) accounts for the conceivable, explicable, and visible "functions of the real mark" (*shixiang yong* 實相用) (*Zhongguan lun shu* T42, no. 1824, p. 124, a14–20). The Chinese term "*busi* 不四," literally "not-four" and here translated as "what is not divided into four [within all four]," describes the indivisible which is the "cohesive body of the real mark" (*shixiang ti* 實相體). Neither of the four can be understood as an independent ontological predication of what reality ultimately is, yet each of the four must be seen in its specific heuristic value and significance.

The cohesive body of the real mark, understood as an indivisible whole, is invisible or hidden in the sense that meaning conveyed by awakened speech evades any fixed or determinate form of predication. If the *catuṣkoṭi* is to be seen as a linguistic symbol for an exhaustive set of alternative predications of reality, the indivisible and invisible body of the real mark—cohesion of the ultimate meaning of awakened speech—is not confined to just one apart from the other three predicative forms, nor is it what goes beyond or separates from all four. Cohesion is what sustains the four distinct alternatives of predication, each of which is the visible form of the body's functioning.

This is analogous to the emptiness that constitutes all differing phenomena which are incessantly changing instances of conditioned arising, each of which points back at the very nature of all, emptiness of an intrinsic nature; or it is like ultimate truth that constitutes all conventional truths; each is capable of guiding the practitioner back to the very source it descends from. The two aspects of the *catuṣkoṭi*, that is, the hidden cohesion of ultimate meaning and its differing functions of predication, imply a sense of circularity which, as will be shown below, accounts for the hermeneutical circle that the mind must enter to accomplish liberation from its own deceptions.

Again, the hidden cohesion of the ultimate meaning can be apprehended in each of the four distinct forms (the visible)—like emptiness in conditioned arising. This can be illustrated on the basis of the previous explanation that the four alternatives in the *catuṣkoṭi* are interdependent or correlative opposites: the first asserts what the second denies and

vice versa, which brings about the third, embracing the entire opposition of the two; the opposite of this is the fourth excluding the first and second; like the first and the second, the third and the fourth are interdependent opposites. As each of the four entails all four, we could say that the invisible and indivisible which constitutes all—the hidden cohesion of all four—is equally embodied by each of the distinct four.

The interlinked relationships make up the "cohesive body of the real mark" (*shixiang ti*) which is not really divisible into four (*busi*), although its functions can and must be classified into four distinct forms. The "real mark" has both a hidden (indistinct) and a visible (distinct) side, inseparably bound up with one another. Its cohesive body cannot be represented by any of its distinct functions alone, and yet the cohesion of "not-four" (*busi*), the indivisibility of the real mark, or the ultimate meaning is wherein all four functions emerge as mutually distinct and visible forms.

The practitioners' awareness of the differentiation and non-duality of the distinct and indistinct sides of the real mark—the "four functions" and the indivisible "not-four"—features the understanding of the ultimate meaning, which can be described in terms of the hermeneutical circle. Distinguishing the two (the distinct and the indivisible) entails realizing non-duality; the same applies in reverse: seeing their non-duality requires differentiating the two. According to Jizang's exposition, entering the hermeneutical circle in such examination of the *catuṣkoṭi* triggers the understanding of ultimate meaning, which requires awareness of the paradox that the mind's intent of separating from its own fabrications entails its clinging to them. This is so, because what is unreal cannot be appropriated, nor can it be dismissed, and yet its existential relevance cannot be denied either. Such awareness culminates in seeing that the ultimate meaning instantiating the nature of reality (real mark or liberation from deception) evades ontological determination.

In his treatise on the *Lotus Sutra* (*Fahua xuanlun* 法華玄論), Jizang explains ontological indeterminacy through the term "indeterminate mark" (*wuding xiang* 無定相), which is just another term for real mark (*shixiang* 實相) and, in this discussion, is directly linked to the *catuṣkoṭi*. According to that treatise, the indeterminate mark or real mark is dynamic, non-excluding, and infinite (*wuliang* 無量), alternately taking shape in any of the four alternatives. Jizang explains:

> Hence, one should know that the [Buddha-]dharma has no determinate mark; it is just due to [the act of] awakened understanding that there is such following [of any of the four]. . . . The one fully aware of this realizes in respect of all four cases that they are [the ultimate meaning of] the Buddhadharma, whereas the one unaware of this, in all four cases, falls prey to the dharma of deception. . . . The one, who accomplishes wisdom in terms of *prajñā* and *upāya*, while studying this *catuṣkoṭi*, is not likely to fall prey to any of the four [false] views [by clinging to them]. 故知: 法無定相, 唯悟是從. . . . 了者, 於四句皆是佛法. 不了者, 四句皆是魔法. . . . 得般若, 方便, 學此四句, 不墮四見. (*Fahua xuanlun* T34, no. 1720, p. 381, a22–b14).

Jizang clearly stresses the ambiguous potential of the *catuṣkoṭi*, as adopting such a method of classifying conceptual forms either manifests the ultimate meaning, or may generate confusion. Most importantly, there is no fifth or even sixth *koṭi* (edge) that goes beyond all four or is within all four, and yet the way in which all four predicate the indeterminate mark—the real mark or ultimate meaning—is to not assert any of the four, nor to deny any of them—not asserting amounts to not denying and vice versa.

Again, to observe this paradox is to see that the ultimate meaning instantiating the nature of reality (real mark) evades ontological determination. Even the view that contradiction is the definite form that truly predicates ultimate reality contradicts Jizang's sense of ontological indeterminacy. Such interpretation entails entanglement in deceptive distinction, clinging to one of the four—the third *koṭi*. The paradox which Jizang reveals is not a contradiction expressing ontological significance. Metaphysics of dialetheism should not be superimposed on Jizang's practice of observing paradoxes in his soteriological approach to exegesis.

Zhiyi's Tiantai view of Nāgārjuna's positive version of the *catuṣkoṭi* is very similar to the Sanlun interpretation. The Tiantai master also emphasizes that the *catuṣkoṭi* represents the last of the four *siddhāntas*, and, similarly, the discussion must take two aspects into account. In his treatise on the *Lotus Sutra* (*Miaofa lianhua jing xuanyi* 妙法蓮華經玄義) and his commentary on the *Vimalakīrti-nirdeśa-sūtra* (*Weimo jing xuanshu* 維摩經玄疏), Zhiyi quotes the *catuṣkoṭi* from the *Da zhi du lun* and expounds the ultimate meaning (*diyi yi*) or *dharmatā* in reference to (1) the explainable (*keshuo* 可説) and (2) the unexplainable or ineffable (*bukeshuo* 不可説; T33, no. 1716, p. 687, a12–22; T38, no. 1777, p. 520, c15–28).

For him, the four alternatives in the *catuṣkoṭi* represent the explainable side (*keshuo*) of the ultimate meaning (*diyi yi*), which comes close to Jizang's "functions of the real mark" (*yong* 用), while the ineffable or unexplainable resembles Jizang's "cohesive body of the real mark" (*ti* 體). Zhiyi further states that those who do not see non-duality in such distinction between the explainable and unexplainable fall prey to their clinging to either one of the four, generating nothing but "*prapañca*" (*xilun*)—entanglement in deceptive speech, while the awakened who integrates the four sees one in four and four in one, without taking either one, or four, or both, or neither as the ultimate meaning. The awakened practitioner understands the "real mark of all dharmas" (*zhufa shixiang* 諸法實相), by observing non-duality while differentiating the explainable and ineffable sides.

Where Jizang's and Zhiyi's interpretations converge is the point that all the alternatives in the *catuṣkoṭi* are interdependent links apart from which the ultimate meaning cannot be realized—there is no fifth, sixth, or any further point; also, any of the four separated from the other three would contradict that meaning. According to both Zhiyi and Jizang, the *catuṣkoṭi* is the conceptual blueprint for doctrinal classification. However, unlike Jizang, Zhiyi seems to relate this specific *catuṣkoṭi* only to the second level of his fourfold classification of doctrines (*huafa sijia* 化法四教). In order to understand the difference between Jizang's and Zhiyi's interpretation, the aspect of doxography (doctrinal classification) must be discussed.

## 5. The Catuṣkoṭi as a Conceptual Blueprint for Tiantai's Doxography

In the *Mohe zhiguan*, Zhiyi's Tiantai teaching explicitly hints at the ambiguous potential of the *catuṣkoṭi*, distinguishing opposite methods of its application: (1) the *catuṣkoṭi* as a proper linguistic means which can convey the ultimate meaning realized by the awakened or liberated mind, or (2) the same linguistic forms in deceptive speech arising from the non-awakened state of mind (T46, no. 1911, p. 68, a29–b57).

According to Tiantai, the first type embraces the perceived form of proper doctrinal discourse in sūtra and śāstra, and, as a consequence of this perception, Zhiyi and his eminent disciple, Guanding 灌頂 (561–632), constantly resort to the *catuṣkoṭi* as an analytical technique of examining all the exegetical issues which they discuss in their own works. In this regard, the *catuṣkoṭi* is dealt with in a twofold way: (1) it is an instrument of conceptual analysis by means of which Tiantai construes and interprets doctrinal contents of Indian Buddhist texts; (2) it is itself an object or topic of examination.

In the *Mohe zhiguan*, Zhiyi first analyzes the *catuṣkoṭi* as it figures in deceptive speech and then expounds how it functions in proper doctrinal discourse; the two steps are connected with one another: he first uncovers the paradox that the intent of separating from the four alternatives entails the reverse—the mind's clinging to them (see Section 6). He further holds that revealing this paradox is the prerequisite for disclosing to the deluded practitioners the salutary side of the *catuṣkoṭi*, namely, its application as a means of conceptual analysis in doctrinal discourse and exegetical discussion. This culminates in establishing Tiantai doxography modeled after the four alternatives. Its taxonomy consists of four levels, classifying and assorting the perceived variety of doctrinal content in sūtra and śāstra:

(1) The first level, called "Tripiṭaka teaching" (*sanzang jiao* 三藏教), holds the position of being (*you* 有) equivalent to the first *koṭi* "everything is real" (*yiqie shi*), and represents Hīnayāna as it appears to Tiantai in the shape of Sarvāstivāda in the *Āgama-sūtras* and

*Abhidharma-śāstras*. Although this view comes very close to Bhāviveka's interpretation of the first *koṭi*, Tiantai stresses that such a form of Hīnayāna does not develop any point of intersection with Mahāyāna.

(2) The second level, called "common teaching" (*tongjiao* 通教), represents the reverse position of non-being (wu 無), equivalent to the second *koṭi* "everything is unreal" (*yiqie feishi*), and reveals what Hīnayāna and Mahāyāna share in common; this encompasses some of the doctrinal contents expounded by the *Prajñāpāramitā*- and *Vaipulya-sūtras* and also indicates similarities with Bhāviveka's view of the second *koṭi*.

(3) The third level, called "distinct teaching" (*biejiao* 別教), is the position of both being and non-being (*feiyou feiwu* 亦有亦無), equivalent to the third *koṭi* of "both real and unreal" (*yi shi yi feishi*). "Distinct teaching" designates the sublime sense of Mahāyāna, excluding Hīnayāna; Tiantai perceives this in the Yogācāra texts and even more obviously in the Tathāgatagarbha texts, as the latter tend to stress the wisdom of both emptiness (*śūnya*) and non-emptiness (*aśūnya*, *bukong* 不空).

(4) The fourth level, called "round/perfect teaching" (*biejiao* 圓教), takes the stance of neither being nor non-being (*feiyou feiwu* 非有非無), equivalent to the fourth *koṭi* of "neither real nor unreal" (*yi shi yi feishi*). The round/perfect teaching embodies circularity, as it is both the source and the goal of the preceding levels. With varying degrees of explicitness, it appears in all Mahāyāna sūtras, but it is said to be most distinctively manifested by the "one [Buddha-]vehicle" (*ekayāna*, *yisheng* 一乘) in the *Lotus Sutra*.

Moreover, "round/perfect" (*yuan* 圓) also indicates the sense of suspending differences, leveling out opposite forms of one-sidedness, and undermining the duality between, or exclusivity of, views, conceptual positions, and mental attitudes; therefore, it also epitomizes the meaning of the middle way (*madhyamaka*, *zhongdao* 中道), often defined as "neither separating nor adhering" (*buli buji* 不離不即). This is to say, the round/perfect level of "neither being nor non-being"—the middle way—does not exclude any of the other three positions; it does not go beyond the preceding three. The round/perfect encloses all of them and is capable of manifesting the sense of "neither being nor non-being" through these differing forms of "being", "non-being", and "both being and non-being". In this fashion, Tiantai tries to show that it neither separates from nor clings to anything, realizing the sense of liberation. Moreover, this view of the *catuṣkoṭi* implies a reinterpretation of Madhyamaka's two truths.

Apocryphal sūtras, composed in China during the fifth century, started to talk about a third truth, that is, the "highest meaning of truth in/as the middle path" (*zhongdao diyi di* 中道第一義諦). Tiantai and Sanlun adopted this notion and construed the concept of the "threefold truth" (*sandi* 三諦). According to Tiantai's round/perfect teaching, this means that separation of the middle from the conventional and the ultimate undermines the very meaning of those two. At the fourth level of doctrine, conventional truth and ultimate truth, respectively, instantiate the highest meaning of truth in the middle path, and each equally displays the ultimate meaning of the Buddhadharma. Each of the three truths embodies all three. To understand this is to simultaneously see one in three and three in one, yet it does not imply the reality of one and that of three.[23] Such an ambiguous and paradoxical structure of the threefold truth is called "inconceivable" (*busiyi*).

Again, the highest level of doctrine presenting the inconceivable sense of threefold truth is the round/perfect teaching (*yuanjiao*). For heuristic reasons, it engenders its own reverse—the lower three levels which can lead the non-awakened practitioners back to the highest. However, in contrast to the inconceivable round/perfect teaching, the lower three of all four levels in Tiantai doxography conceptualize the ultimate meaning of Buddhist doctrine, generating instructive yet incomplete notions of it. In this way, those lower teachings no longer maintain mutual integration of the three aspects that constitute truth in such cohesion. Only the manner in which the round/perfect teaching exhibits the middle path as the all-embracing Buddha nature (nature of awakening, *foxing* 佛性) in both conventional truth and ultimate truth features the complete sense of the threefold truth—the Tiantai sense of inconceivable.

Although Tiantai master Zhiyi construes his taxonomy of Buddhist doctrine on the methodological basis of the *catuṣkoṭi*, at this point, he reverses the course of discussion: the *catuṣkoṭi* turns into an object of examination and therefore must be reinterpreted from the viewpoint of all four doctrinal levels (*Mohe zhiguan* T46, no. 1911, p. 73, b25–p. 75, b27). The *catuṣkoṭi* realized in its inconceivable sense due to the round/perfect teaching does not equal those meanings which itembodies when it is seen from the vantage points of each of the three conceptualizing levels. The same applies to all doctrines transmitted in the Indian Buddhist texts—according to Tiantai. All notions mentioned in sūtra and śāstra—such as the two truths, conditioned arising etc—must be examined from the vantage point of this fourfold classification, called the "four teachings of the transforming dharma" (*huafa sijiao*).

Hence, each of the four levels of teaching looks at the same linguistic form of the *catuṣkoṭi* in a different way. Depending on the doctrinal level of discussion, each of the four ("being", "non-being", "both being and non-being", "neither being nor non-being") changes its meaning four times. Despite such content-related amplification of each of the four expressions, the inner relations between all 16 modifications together are evident. Hence, assorted in this pattern of interdependence and mutual constitution, all of them are unified as a systematic whole. In this fashion, they are believed to systematically evince the hidden and inner coherence of the doctrinal diversity in sūtra and śāstra. Zhiyi quotes extensively from all Indian Buddhist texts, to find canonical evidence for each of the 16 views which he considers as that whichsystematically summarizes, epitomizes, and encompasses the entire content of doctrinal discourse in the transmitted canon of sūtra and śāstra. Aside from that hermeneutical purpose, the intended soteriological effect of this classification and expansion is that the ascension of a certain doctrinal topic from the lower conceptualizing levels to the highest inconceivable, that is, the integration of its conceptualization into the inconceivable sense, becomes intelligible to practitioners.

Zhiyi's treatment of the *catuṣkoṭi* pursues the hermeneutical goal of highlighting the inner unity of the numerous and various doctrines in the translated texts from India. This coincides with his soteriological intent of showing that the various textual representations of the Buddhadharma are congruent with the ultimate meaning of liberation. In this way, he demonstrates that the hidden sense of coherence—for him the highest meaning of awakening—ultimately is independent from linguistic signification, and yet does not really separate from speech and concept. For him, the textual corpus of the Buddhist canon embodies precisely this. Again, Tiantai's doxographical appropriation of the *catuṣkoṭi* mirrors how the hermeneutical project concurs with the soteriological intent: to deconstruct the perceived duality of doctrinal conceptualization and inconceivable liberation is to integrate the ineffable—the ultimate meaning—with discourse in sūtra and śāstra. Nāgārjuna's *catuṣkoṭi* is a major factor in this project.

## 6. The Catuṣkoṭi as a Symbol for Deception

However, how does the same *catuṣkoṭi* figure as a deceptive form of speech characterizing the deluded or non-awakened mind? To make the deceptive effects of the *catuṣkoṭi* transparent, Zhiyi develops the Tiantai practice of contemplating the mind (*guanxin* 觀心)—a practice of introspection that detects all the blindspots which the non-awakened mind itself creates and falls prey to. The contemplating mind comes to see that certain types of distinctions are the source of mistaking the referents of our speech for independently existing entities. Tiantai commences with contemplating emptiness (*kongguan* 空觀) which first deconstructs the assumption of an intrinsic nature immanent to things and then unveils the paradox that the intent of separating from the distinct forms of the *catuṣkoṭi* entails the reverse—the mind's clinging to deceptive distinctions.

This contemplation is designed for the common or non-awakened persons (*fanfu* 凡夫) who not only fail to liberate their minds from deceptive construction, but also increase the same. Taking each of the four alternatives as a logically independent or exclusive predication, they end up construing even a *catuṣkoṭi* of the *catuṣkoṭi*. Zhiyi classifies such unrestrained proliferation of non-awakened speech (*prapañca, xilu*) into [1] the simple

*catuṣkoṭi* (*dan siju* 單四句) which construes ontological views about being, non-being, both, neither, [2] the double *catuṣkoṭi* (*fu siju* 複四句), [3] the multifarious *catuṣkoṭi* (*juzu siju* 具足 四句), and [4] the dissolution of speech (*jueyan* 絶言), also called absence of speech (*wuyan* 無言) (*Mohe zhiguan* T46, no. 1911, p. 62, b8–10). This fourth position represents the mind's intent of realizing complete silence, separatefrom the preceding three types of the *catuṣkoṭi*. Compared to the simple *catuṣkoṭi*, the two types of the double and multifarious *catuṣkoṭi* increase the complexity of each of the four alternatives.

The double *catuṣkoṭi* (*fu siju*) might emerge, when the four alternatives of the simple *catuṣkoṭi* are rejected without realizing the sense of ontological indeterminacy. More complex forms of metaphysical confusion arise from this, embracing the following four alternatives: (1) being of being, being of non-being (有有, 有無); (2) non-being of being, non-being of non-being (無有, 無無); (3) both being of being/non-being and non-being of being/non-being (亦有有無, 亦無有無); (4) neither being of being/non-being, nor non-being of being/non-being (非有有無, 非無有無, T46, no. 1911, p. 62, c12–13). Seeing that the first and second *koṭi* in the simple *catuṣkoṭi* are correlative opposites mutually constituting each other, the deluded mind construes the double *catuṣkoṭi* and holds that "being" must represent being of both being and non-being, while the opposite "non-being" means non-being of both being and non-being; the same applies to the third (including the preceding two) and to the fourth (excluding them).

Zhiyi further explains that, if one intends to escape from the entanglements in the double *catuṣkoṭi*, but still continues to lack an awareness of ontological indeterminacy, one might fall prey to the multifarious *catuṣkoṭi*, which entails one's involvement with even more complex forms of confusion. When holding the first position "being" in the multifarious *catuṣkoṭi*, the deluded mind observes that all four alternatives are mutually constitutive to one another, therefore "being" includes four aspects: being of being (*you you* 有有), being of non-being (*you wu* 有無), being of both (*you yiyou yiwu* 有亦有亦無), and being of neither (*you feiyou feiwu* 有非有非無); the same applies to the opposite "non-being", to the third, and to the fourth; each of the four includes four (*Mohe zhiguan* T46, no. 1911, p. 62, c16–23).

Most importantly, these 16 deluded views are not equivalent to the 16-fold sense of the four levels of teaching in Zhiyi's doxography of the Buddhadharma. Although examining the *catuṣkoṭi* from different perspectives, each of these four taxonomic levels helps achieve awareness of ontological indeterminacy which eventually dissolves all metaphysical dogmatism, whereas each of the four positions of the multifarious *catuṣkoṭi*, to the contrary, strengthens the grip on one specific ontological view excluding the others, which exacerbates metaphysical confusion.

Even if one's mind rejects all three types of the simple, double, and multifarious *catuṣkoṭi*, adhering only to the fourth position of complete dissolution of speech, the mind locks itself into a deceptive form of silence. Silence as the result of the mind's intention of eliminating speech has not really resolved the mind's clinging to deceptive dualities; such silence does not differ from speech. The mind is still captured in the circular entanglement of deceptive distinctions which it has construed due to its misuse of the conceptual forms of the *catuṣkoṭi*. Such dualistic silence is not identical to the silence described in the *Vimalakīrti-nirdeśa-sūtra*, which is non-dual in the sense of neither excluding speech nor silence. Therefore, the exclusive sense of silence is classified as the fourth position in the deceptive version of the *catuṣkoṭi*. In the *Mohe zhiguan*, Zhiyi explains the whole structure:

> [When contemplating the mind, one might realize that] the four alternatives are false/provisional [forms of designation], they are deceptive and unreal. The true principle [of non-arising] goes beyond word and speech, dissolving all four alternatives, which indeed is non-arising. Although one might then say [such insight] is what makes one overcome the four alternatives, in fact one has not overcome them at all. Briefly, there are three types of [intentions] of going beyond the four alternatives: first, [going beyond] the simple; second, [going beyond] the double; third, [going beyond] the multifarious [*catuṣkoṭi*]. If one

says that by virtue of true principle one goes beyond speech, then this is only overcoming the simple *catuṣkoṭi*, but one has not overcome the second alternative of the double *catuṣkoṭi* [equivalent to non-being of both being and non-being], nor has one overcome the first of the multifarious *catuṣkoṭi* [equivalent to being of neither being nor non-being]. One should know: the net of deceptive views is closely knotted, it is difficult to escape from it. . . . One might say again: Stepping beyond the simple, double, and multifarious *catuṣkoṭi* and overcoming them, this is wherein speech and debate come to an end, circulating thoughts extinguish, and obliteration and purification persist, as it is the path of non-arising and dissolution of speech. The one who calculates in this way yet lapses into the [deluded] view about the ineffable and dissolution of speech. What has this to do with the proper path? This one only says 'dissolution of speech', but when one's speech ends, one's speech is not dissolved. Why is it so? One expounds dissolution depending on non-dissolution with the result that dissolution yet implies dependency. As long as the sense of interdependent opposites arises, one should not speak of dissolution. It is as if one tries to escape from empty space, but how is it possible to evade true principle [which embraces all]? 四句皆假, 虛妄不實. 理在言外, 絕於四句, 乃是無生. 謂出四句, 實不出也. 略有三種四句外: 一單, 二複, 三具足. 若謂理在言外者, 乃是出單四句外, 不出複見第二句, 亦不出具足見初句. 故知: 見網蒙密, 難可得出. . . . 又復言: 出單, 複, 具足四句之外, 言語道斷, 心行處滅, 泯然清淨, 即是無生絕言之道. 如此計者, 還是不可説絕言之見. 何關正道？徒謂絕言, 言終不絕. 何以故? 待不絕而論絕, 絕還是待. 待對得起, 不應言絕. 如避虛空, 豈有免理! (*Mohe zhiguan* T46, no. 1911, p. 66, c19–p. 67, a5).[24]

Overall, Zhiyi's purpose of construing the fourfold classification of the deceptive *catuṣkoṭi* is to highlight all the blindspots which impede the deluded mind. Therefore, he uncovers the paradox that the intent of excluding the four distinct forms intensifies the mind's involvement with deceptive distinction, producing new types of the *catuṣkoṭi* even more complex than those it tries to eliminate.

As long as the mind does not realize that its generation of delusive views functions like the circular operation of a closed system, it continues to confuse its own fabrications with real entities, erroneously holding that the objects of its epistemic–propositional references pertain to a world of independent things. The mind's circular entanglement in its own metaphysical deceptions is like a self-contained system. Hence, enacting such features of the deluded mind, the scheme itself must be self-referential. It subsumes various types of the *catuṣkoṭi* in non-awakened speech and silence under the same pattern of the *catuṣkoṭi*, assorting the gradated levels of the mind's entanglement in metaphysical constructions which all deceptive. The fourfold *catuṣkoṭi* is a symbol for the mind's unawareness of its circular entanglement in self-induced deception.

However, Tiantai also believes that it is the mind's recurrent experience of falling prey to its own deceptions which makes it realize that there is nothing it can really hold on, neither within nor apart from its circulating in metaphysical dogmatism. Then, it really comes to see that the objects of its epistemic–propositional references are not independently existing entities, which amounts to seeing their real nature, yet being fully aware of ontological indeterminacy. Again, this is neither separating from anything, nor clinging to anything, nor asserting anything, nor denying anything, as what is unreal cannot be appropriated, nor dispensed with.

Yet, to see what unreality actually and really is means to apprehend its existential relevance—veiled and concealed in respect to what it is in actuality, unreality is deceptive and leads one astray, which brings suffering. However, to reveal and expose its real nature is, in effect, to explore the path to liberation from deception. Aware of what it truly is, the deceptiveness of unreality is instructive. Unreality retains its heuristic significance due to its deceptiveness. Therefore, the same *catuṣkoṭi* also functions as a symbol of Buddhist doctrine, opening up the path to liberation and awakening.

### 7. Concluding Remarks

The article aims to show that, in the Chinese appropriation of the Indian *catuṣkoṭi*, the ambiguous potential of this method of classifying conceptual–linguistic forms was a major factor which shaped the development of Buddhist debate in East Asia. The *catuṣkoṭi* was considered as that which might either inspire awareness of ontological indeterminacy or entail the reverse effect of generating confusion over metaphysical dogma. In China, the *catuṣkoṭi* served the purpose of classifying both doctrinal discourse in the transmission of the canonical word and non-Buddhist metaphysics considered to be deceptive.

Hence, Chinese masters hold that, in order to avoid mistaking the *catuṣkoṭi* in a way which entails further clinging to deceptive distinction, its ambiguity must be perceived when studying Indian sūtras and śāstras. Therefore, in the *Mohe zhiguan*, Zhiyi unfolds the inner connection of those opposite aspects, assorting ten inter-connected types of *catuṣkoṭi* (*shizhong siju* 十種四句). Jizang partially uses the same expressions that Zhiyi mentions; however, his list of the ten types of *catuṣkoṭi* (*shizhong simen* 十種四門) seems to follow a dialectical pattern, culminating in the sublation of the four alternatives, called "suspending (sublating) four" (*juesi* 絶四).[25] Nonetheless, according to Jizang, "suspending four" is preciselywhat validates the use of the four alternatives in the sense of the Dharma. Despite the differences in Zhiyi's and Jizang's approaches, the sense of ambiguity remains the same.

Moreover, Jizang's view also seems to come close to what Zhiyi, in the *Mohe zhiguan*, calls the "inseparability of deconstruction and construction" (*jipo jili* 即破即立)—the deconstruction of deception is the construction of doctrinal discourse or vice versa. Deconstruction of deception in the practice of contemplating emptiness engenders doctrinal discourse which in the shape of sūtra and śāstra provides the guideline that the practitioners must pursue. In such circularity, doctrine and contemplation are mutually complementary (*jiaoguan xiangzi* 教觀相資).

Whereas Zhiyi's concern in the *Mohe zhiguan* consists of linking the structural pattern of the *catuṣkoṭi* to his method of mind-contemplation (*guanxin*) in tandem with his doxography (*panjiao*), Jizang applies the *catuṣkoṭi* as a strategem in linguistic pragmatics to make practitioners' aware of certain paradoxes when examining the sense of non-duality (*buer* 不二), which debunks deceptive distinction via differentiation of the two truths. In his *Dasheng xuanlun* 大乘玄論, *Shier men lun shu* 十二門論疏, and the *Erdi yi* 二諦義, Jizang presents his respective Sanlun teachings of three or four levels of two truths (*sichong erdi* 四重二諦) guiding that sense of non-duality. The *Dasheng xuanlun* further expands on the same scheme in terms of doxography (Chen 2013), but all three works discuss the *catuṣkoṭi* in combination with the two truths.

Overall, the Tiantai and the Sanlun versions can be subsumed under what I call hermeneutics of deconstruction, because the deconstructive goal of the two as well as their motif of taking the entire canonical transmission from India into account seems to be the same.

Furthermore, Tiantai and Huayan are similar in the specific respect that the two develop their doxographies on the basis of the *catuṣkoṭi* and, in turn, analyze the same *catuṣkoṭi* from the viewpoint of their doxographies. This entails a type of circularity which the two Chinese schools share in common: the *catuṣkoṭi* figures as both the tool and object of their exegetical examinations. They also discuss their doxography in the same circular manner: the taxonomy of doctrine appears to be both the method and the topic of their discussions. Such treatment of the *catuṣkoṭi* in the context of doxography slightly differs from Jizang's Sanlun approach, although all three Chinese schools seem to agree that it must be a certain sense of circularity based on which soteriological theory in Buddhism can solve the question of how transformation from a non-awakened state of being into the opposite becomes possible.

**Funding:** This research received no external funding.

**Conflicts of Interest:** The author declares no conflict of interest.

## Abbreviation

T = Taishō shinshū daizōkyō 大正新脩大藏經 [Buddhist Canon Compiled during the Taishō Era (1912–26)]. See Primary Sources, Takakusu and Watanabe, et al., eds.

## Notes

[1] The Chinese term "*jiaming*" (false/provisional name) accounts for the Sanskrit "*prajñapti*," which is a causative participle of the verb root "*jñā*" (to know) and can be understood as "what is made known, or to be shown". The Chinese "*jia* 假" literally means "to borrow, pretend, as if, false, provisional". "Borrowed names" (*jiaming*) represent what is to be shown; however, more explicitly than the Sanskrit "*prajñapti*", the Chinese semantics of "*jiaming*" convey the sense of "falsehood," which Indian *Prajñāpāramitā-sūtras* and Madhyamaka texts indeed imply, as, according to their doctrine, no object of any linguistic signification represents an entity sustained by an irreducible or real core.

[2] For instance, in the influential *Lotus Sutra*, the Buddha announces that his various performances and teachings are not really what they appear to be; even his extinction into nirvāṇa is unreal. Yet, such unreality is not misleading; on the contrary, it points back to truth and therefore is deemed to be instructive.

[3] In his commentary on Nāgārjuna's *Kārikā*, the *Zhongguan lun shu* 中觀論疏 and some other works, Jizang calls this "*jiewang zhiwang* 借妄止妄", which literally means "to borrow [instructive] falsehood in order to cease [deceptive] falsehood" (T42, no. 1824, p. 18, c28–29).

[4] The notion of "non-duality" or "inconceivable oneness of root and traces" (*benji busiyiyi* 本跡不思議一), inspired by Guo Xiang's郭象 (252–312) commentary on the Daoist work *Zhuangzi* 莊子 (Kantor 2019b, pp. 103–35), appears for the first time in Sengzhao's 僧肇 (374–414) commentary to the *Vimalakīrti-nirdeśa-sūtra*, which also includes Kumārajīva's and Zhu Daosheng's 竺道生 (360–434) annotations, *Zhu Weimo jing* 注維摩詰經 (T38, no. 1775, p. 327, b4–5). Tiantai master, Zhiyi, and Sanlun master, Jizang, adopted the binary "root and traces" (*benji* 本跡) from Sengzhao and used it as an exegetical tool in their commentaries on Indian sūtras.

[5] A similar thesis has been articulated by Young (2015, pp. 111–52).

[6] Contemporary studies on Nāgārjuna's *catuṣkoṭi* generally deal with the question of how the logical structure of the four types of predication can be described by means of formalized language. Raju (1954, pp. 694–713), Robinson (1967, pp. 291–308), and Jayatilleke (1967, pp. 69–83) represent the earlier attempts of applying propositional logic, while Nakamura (1954, pp. 223–31) resorted to the means of symbolic logic. Matilal (1971, pp. 43–89) discussed the same topic in the frameworks of logic, epistemology, and grammar in Indian philosophy, thereby focusing on speech act theories. Ruegg's (1977, pp. 1–77) lengthy article on the metaphysical implications of the *catuṣkoṭi* in Mahāyāna includes the first comprehensive review of Western studies. This was followed by Wood's (1994) nihilistic interpretation of Nāgārjuna's philosophy, criticizing Jayatilleke, Matilal, and Seyfort Ruegg for holding a view incongruent with the meaning of *śūnyatā*. His thesis of Madhyamaka nihilism was in turn challenged by Priest and Garfield (2002, pp. 249–70), and Deguchi et al. (2008, 2013) who developed their thesis of true contradictions in Madhyamaka thought. Priest's recent monograph (2018) understands the differing applications of the *catuṣkoṭi* as an evolution of logical investigation in the Buddhist traditions across India and China. However, his discussion omits the Tiantai view of the *catuṣkoṭi*, although Tiantai elaborates on this classifying scheme much more than other Chinese Buddhists do. All the varying attempts of applying formal language in the approach to the *catuṣkoṭi* developed huge distances to the interpreted texts, neglecting the conceptual context with other important doctrines. Questions arise, such as whether those approaches are helpful in understanding Buddhist texts; do they pinpoint problems which the primary sources have not been aware of; do the suggested solutions of contemporary scholars improve the arguments in the ancient sources? Or, conversely, do these discussions help advance the study in logic and analytic philosophy which has developed independently from Buddhist thought? Moreover, must or should Buddhist thought in pre-modern China be evaluated according to what contemporary strands of academic philosophy consider as relevant in order to count as philosophy (van Norden 2017; van Norden and Garfield 2016; Priest 2018) in the first place, given the fact that such strands are as temporary as previous paradigms in philosophical thought were (Ziporyn 2019; Kantor 2019b; Stepien 2019)? For a similar concern, see Westerhoff (2020, pp. 965–73) and footnote 18.

[7] For instance, Zhiyi explicitly says this in the *Mohe zhiguan* (T46, no. 1911, p. 68, a29–b5). Similar remarks are made in Jizang's Sanlun and Fazang's Huayan teachings, *Huayan yisheng jiaoyi fenqi zhang* 華嚴一乘教義分齊章 (T45, no. 1866, p. 499, c23–p. 500, a1).

[8] The Tibetan tradition transmits the *Mūlamadhyamaka-vṛtty-akutobhayā*, a commentary to the *Kārikā*(verses),which some contemporary scholars ascribe to Nāgārjuna himself. However, the Chinese Mādhyamikas had no knowledge about this text. Together with six commentaries, each composed by a different Indian author, the *Kārikā* has been transmitted in one Sanskrit, three Tibetan, and three Chinese versions. Bhāviveka's *Prajñāpradīpa* exists in both Chinese and Tibetan, but not in Sanskrit. The oldest and, in Chinese sectarian Buddhism, most influential version is Kumārajīva's *Zhong lun*, which does not mean that it represents Nāgārjuna's original thoughts. From a methodological point of view, the discussion about Kumārajīva's transmission and its adoption by Chinese Sanlun, Tiantai, and other masters must be separated from the question of what Nāgārjuna's ideas might have been in their own right. The article deals only with the first but not with the second question, as indicated by this section title. The same applies to the next section. Section two and three examine Nāgārjuna's, Piṇgala's and Bhāviveka's views of the

tetralemma, with special focus on how the textual tradition in China has transmitted them. I agree with the reviewer that it would have required another approach to those three thinkers, if the significance of their thoughts were to be discussed solely in the context of Indian philosophy; in that case, one would have needed to analyze both the differences between and commonalities among all transmissions. An initial step into this direction has been made by Mitsuyoshi. His translations of the *Kārikā* versions from Sanskrit, Tibetan, and Chinese into modern Japanese show that those parts of Kumārajīva's work, which are relevant to this article (that is, the tetralemma and its associated doctrines), do not greatly differ from Candrakīrti's transmission in Sanskrit, the Tibetan \**Akutobhayā*, and Bhāviveka's and Sthiramati's texts preserved in Chinese; see Mitsuyoshi (1985).

[9] On negativity in Nāgārjuna's use of the *catuṣkoṭi*, see Ruegg (1977), Wood (1994), Ng (1993), Westerhoff (2005), and Zhang (2016).

[10] Commitment to "*pratītyasamutpāda*" (*yuanqi*, conditioned arising) is what distinguishes all Buddhists from non-Buddhists, from a Buddhist point of view. Transformation initiated by insight, practice, and cultivation implies causal change. Without such presupposition, none of the practitioners' efforts could give rise to the fruit of liberation. Conditioned arising implies neither discontinuity nor permanence of entities, because permanence excludes change, and discontinuity denies causal effects. In the Chinese Madhyamaka tradition, "neither permanence nor discontinuity" (*buchang buduan* 不常不斷) werethe epitome of Buddhist doctrine, that is, conditioned arising whose true nature is emptiness, whereas positions holding any sense of permance, or the opposite, discontinuity, were regarded as "non-Buddhists" (*fei fofa* 非佛法), as they lack commitment to "*pratītyasamutpāda*" (arising).

[11] Zhiyi's *Mohe zhiguan* makes a similar argument (T46, no. 1911, p. 67, b6–c20), explaining that "non-arising" (*busheng* 不生) in the *Kārikā* also means "not non-arising" (*bu busheng* 不不生). However, as Jizang indicates (T42, no. 1824, p. 28, c8–19), this thought might originate in the *Da zhi du lun* (T25, no. 1509, p. 112, c15–22), traditionally ascribed to Nāgārjuna and said to have been translated by Kumārajīva.

[12] For a more critical review on Westerhoff, see Priest (2018, p. 22), who claims that "*prasajya*" essentially is predicate negation and "*paryudāsa*" sentential negation, rejecting the view that there are two types of negation related to Nāgārjuna's *catuṣkoṭi*. For a reply to Priest, see Westerhoff (2020, pp. 965–74), who basically reasserts his previous view. I believe that Westerhoff's interpretation of "*paryudāsa*" as "presupposition negation" represents only one possible way of understanding this term. Usually translated as "implicative negation," "*paryudāsa*" might perhaps be seen as a factor which constitutes mutuality between opposites. The point is not how Nāgārjuna would have used that Sanskrit term, which he never did, but rather the fact that his application of the *catuṣkoṭi* implies the distinction between two types of negation. This section mostly deals with negation as a constitutive factor for correlative opposites typical of the three types of the tetralemma, drawn from the *Zhong lun* as examples to be discussed.

[13] Some contemporary scholars might not agree with this; however, the three instances of the tetralemma from the *Zhong lun*, discussed in this section, manifest such features. Moreover, it is a fact that such a view is prevalent in the Chinese transmission. Bhāviveka's and Piṅgala's interpretations of the positive application of the tetralemma obviously imply that meaning (see Section 3). Chinese Tiantai and Sanlun masters certainly follow the same understanding: the second alternative denies the first, like the fourth is the negation of the third—these are the two internal negations (see the *Mohe zhiguan* T46, no. 1911, p. 62, b30–c23, but also Sections 3, 4 and 6). For example, the first ([1] emptiness) and the third ([3] both emptiness and non-emptiness) are considered as the two assertive alternatives, while the second ([2] non-emptiness), denying the first, and the fourth ([4] neither emptiness nor non-emptiness), negating the third, are seen as the two negative alternatives (internal negations). The relationship between the third and fourth repeats the one between the first and the second, but in terms of content, the third includes the precedingtwo alternatives, while its opposite, the fourth, excludes the two. In the translated literature from India, this approach is also employed by the *Da zhi du lun* (T25, no. 1509, p. 260, a1–4), which explains what "all dharmas" (all things) assorted in the form of the four alternatives means: "'All dharmas' means: [1] being, [2] non-being, [3] both being and non-being, [4] neither being nor non-being; [1] emptiness, [2] non-emptiness, [3] both emptiness and non-emptiness, [4] neither emptiness nor non-emptiness; [1] arising, [2] perishing, [3] both arising and perishing, [4] neither arising nor perishing; [1] non-arising and non-perishing, [2] not non-arising and not non-perishing, [3] both non-arising/non-perishing and not non-arising/not non-perishing, [4] neither non-arising/non-perishing nor not non-arising/not non-perishing. 一切法, 所謂: 有法, 無法, 亦有亦無法, 非有非無法, 空法, 不空法, 空不空法, 非空非不空法, 生法, 滅法, 生滅法, 非生非滅法, [1] 不生不滅法, [2] 非不生不滅法, [3] 不生不滅亦非不生非不滅法, [4] 非不生非不滅亦非不不生亦非不不滅法." Zhiyi's Tiantai and Jizang's Sanlun discussions of the *catuṣkoṭi* are certainly inspired by the *Da zhi du lun* (see Sections 4 and 6).

[14] Again, this is explicitly outlined in both Zhiyi's *Mohe zhiguan* (T46, no. 1911, p. 82, c2–6) and Jizang's *Zhongguan lun shu* (T42, no. 1824, p. 11, a28–b20). The term "*xiangdai* 相待" (interdependence, mutuality, correlative opposites) occurs in all texts referencing Madhyamaka sources. The verses in the *Zhong lun*, Kumārajīva's translation of the *Kārikā*, use this term only once in chapter 6, where it corresponds to the Sanskrit "*apekṣau parasparam*" (the two are mutually dependent; Siderits and Katsura 2013, p. 67) and Mitsuyoshi (1985, pp. 152–53). In Piṅgala's commentary, the Chinese term occurs 20 times. In his commentary to the *Vimalakīrti-nirdeśa-sūtra*, Kumārajīva's disciple, Sengzhao (僧肇, 374–414), defines it in a way which is also typical of Jizang's and Zhiyi's understanding; Sengzhao explains: "All dharmas arise due to mutual dependency, which is like 'long' and 'short' that shape each other by virtue of contrast. 諸法相待生, 猶長短比而形也," (T38, no. 1775, p. 346, b25–26).

[15] The *Zhong lun* states: "All Buddhas rely on the two truths to expound the dharma [Buddhist doctrine] to the multitude of sentient beings; these are, first, the conventional truth and, second, the ultimate truth. The one, who does not know to distinguish between the two truths, does not understand the true and real meaning in the deep Buddhadharma. Without relying on the conventional

truth, the ultimate truth cannot be apprehended. Without apprehending the ultimate truth, nirvāṇa cannot be accomplished. 諸佛依二諦, 為眾生說法, 一以世俗諦, 二第一義諦. 若人不能知, 分別於二諦, 則於深佛法, 不知真實義. 若不依俗諦, 不得第一義, 不得第一義, 則不得涅槃." (*Zhong lun*, T30, no. 1564, p. 32, c16–p. 33, a3).

16   For the meaning of "ontological indeterminacy", see Ho (2021), Kantor (2019c), and Ziporyn (2016).

17   Resorting to the theory of para-consistent logic, Deguchi et al. (2008, 2013) treat Nāgārjuna's two truths as an instance of dialetheism, that is, certain types of propositions are both true and false at the same time and in the same respect, signifying true contradictions. DGP perceive such true contradictions in a selection of differing Mahāyāna works (Deguchi et al. 2008). Whether Nāgārjuna's paradoxical differentiation of the two truths should be called "dialetheism" is another question, but Mādhyamikas definitely practice the observation of paradoxes. To gain insight into ontological indeterminacy requires awareness of the paradox which inevitably arises when abstinence from any ontological commitment tends to generate the reverse. Unlike DGP, I do not think that paradoxical discourse in Madhyamaka expresses a certain ontological view or metaphysical position. Determining the nature of reality in terms of true contradictions is tantamount to making an ontological commitment, which entails a claim of being, which Madhyamaka tries to deconstruct. Particularly, Tiantai's critique of the mistaken *catuṣkoṭi* denies "being of both being and non-being" (*you yiyou yiwu* 有亦有亦無, see Section 6). Moreover, ontological indeterminacy in Madhyamaka thought requires the distinction between contradiction and paradox. Paradoxical propositions in Madhyamaka do not imply opposite logical values in the same respect. According to Tiantai and Sanlun, all four positions of the tetralemma can be either true or false, depending on the manner and context in which they are used. The deceptive form can become instructive, while the *catuṣkoṭi* properly applied in doctrinal discourse retains potential for deception (see Sections 5 and 6). Its ambiguous potential is ineradicable, and its paradoxical effects are not contradictions. As a clarifying response to one of the reviewers: Another critique of Priest's (2018) dialetheist interpretation of the *catuṣkoṭi* comes from Westerhoff (2020, pp. 965–73). Unlike Priest, Westerhoff believes: "we should be reluctant to assume that different instances of the *catuṣkoṭi* always express the same logical form" (2020, p. 967). Westerhoff also argues that the negative element of the third position in the tetralemma expresses "presupposition failure," an interpretation which does not necessitate the recourse to the theory of true contradictions. However, the third position in all three instances of the *catuṣkoṭi* discussed in this article does not express "presupposition failure", nor does it represent a "true contradiction". The third *koṭi* rather highlights the interdependency of the preceding two *koṭis* which are opposite (see Piṇgala's and Bhāviveka's views in Section 3; the sense of correlative opposites in the third position of the tetralemma is not equivalent with Priest's thesis of true contradictions). My concerns are different from Westerhoff's critique, as they are directed against Priest's thesis that paradoxical discourse in Madhyamaka expresses a certain metaphysical position. Madhyamaka's ontological indeterminacy is incompatible with dialetheist metaphysics.

18   Siderits and Katsura (2013, p. 200) translate from Candrakīrti's Sanskrit version: "All is real, or all is unreal, all is both real and unreal, all is neither unreal nor real; this is the graded teaching of the Buddha".

19   The *Da zhi du lun* says: "It is like the four streams which have different names, before they merge into the great ocean; after flowing into the ocean, there are no differences. . . . Within the realm of conventional truth there are differences, ultimate truth is devoid of differences. 譬如四河未會大海則有別名, 既入大海則無差別; . . . 世俗諦中有差別, 第一義諦則無分別". (T25, no. 1509, p. 611, c9–11).

20   Whether this interpretation of the account transmitted in Chinese is congruent with that which Bhāviveka, as an authentic Indian philosopher, would have thought of is not a topic that in the limited scope of the article can be dealt with. A similar type of *catuṣkoṭi*, interpreted in a way that comes close to the view presented in this discussion, seems to be mentioned in the Sanskrit version of Bhāviveka's *Madhyamakahṛdaya-kārikās*, see Eckel (1992, pp. 34–37).

21   Bhāviveka states: "As to the inner and the external realms, such as all types of entrances and physical forms, these [elements] are real, because, from the vantage point of the conventional truth, they are said not to be inversions. From the vantage point of the ultimate meaning, the inner and the external realms are that which arises due to conditions, fabricated like illusions wherein no real core can be attained, they are not like what they appear to be, therefore [it is said that] everything is unreal. As the two truths are correlatively dependent, [it is said that] this is both real and unreal. Once the practitioners realize the fruit [of their cultivations] and apprehend in all things what is true and real without conceptual differentiations, they do not perceive [any duality of] real and unreal, therefore it is said 'neither real nor unreal'. 内外諸入色等境界, 依世諦法説不顛倒, 一切皆實。第一義中内外入等, 從緣而起, 如幻所作體不可得, 不如其所見故, 一切不實. 二諦相待故, 亦實亦不實. 修行者證果時, 於一切法得真實無分別故, 不見實與不實, 是故説非實非不實". (T30, no. 1566, p. 108, a10–11).

22   "Cohesive body and functions" is cognate with "root and traces," for some contemporary views of its background, see: Muller (2016); Kwon and Woo (2019).

23   For a deeper discussion of this, see Ng (1993, pp. 124–53), Swanson (2018, pp. 115–57), and Ziporyn (2016, pp. 36–89).

24   Swanson's translation of this part in the *Mohe zhiguan* and his understanding of the whole passage on the *catuṣkoṭi* differ from mine (Swanson 2018, p. 900).

25   In Jizang's two treatises on the *Vimalakīrti-nirdeśa-sūtra*, the *Jingming xuanlun* 淨名玄論 (T38, no. 1780, p. 853, a3–p. 859, a27, and T38, no. 1780, p. 857, b20–p. 859, a14) and the *Weimo jing yishu* 維摩經義疏 (T38, no. 1781, p. 912, b9–p. 913, b10), we find a list of ten types of *catuṣkoṭi* which contains those three which are also mentioned in the Zhiyi's *Mohe zhiguan*—the simple *catuṣkoṭi*, the double *catuṣkoṭi*, and the repetitive *catuṣkoṭi* (*chongfu siju* 重複四句). Whether these terms, commonly used by Sanlun und

Tiantai, originate from Zhiyi or Jizang is hard to decide. Recent scholarship in Japan (Hirai 1985) and (Kanno 2007) hold that Zhiyi's disciple Guanding posthumously compiled his mentor's works and most probably borrowed from Zhiyi's younger contemporary, Jizang. However, this thesis is controversially discussed; the reverse might also be possible (Guo 2017).

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
