# Peer review of "Doxographical Appropriation of Nāgārjuna’s Catuṣkoṭi in Chinese Sanlun and Tiantai Thought"

_religions, doi:10.3390/rel12110912_

Round 1
Reviewer 1 Report
I tried very hard to find something to critique in this article, but it is very solid scholarship, well argued, well researched, and so forth, and I have no relevant suggestions for improvement in content or style.
Although I did not read it with copy-editing in mind, here are some very minor editorial suggestions:
The author uses the term Lotus-sūtra, but since “Lotus” is already a translation, I suggest using the more common “Lotus Sutra”.
In note 6, third line from the bottom, there is an extraneous comma: “first place, (Norden” should be “first place (Norden)”
In note 10, the parentheses are not necessary for the references: “see (Seyfort)” can be “see Seyfort” and “2016).” can be “”2016.”
I don’t think it is necessary to repeat textual details after they have already been given at first mention, e.g. repeating “*Mahaprajnaparamitopadesa” at line 320.
There should be some layout indication for blocks of quotes, such as a short left-side indent, for passages such as lines 428-444, lines 466-491, lines 691-699, and lines 921-949.
On line 559, “got lost” is an awkward English phrase, perhaps change to “is not extant”?
Note 23: the author politely comments that his translation and understanding of the Mohezhiguan passage is different from that of Swanson (2018); perhaps it would be useful to the reader to give an example or two of specific ways that there are differences in interpretation?
Line 980: “Conclusive Remarks” should be “Concluding Remarks”?
Line 997: “Dispite” should be “Despite”
Line 1111: “NG” should be “Ng”
Line 1117: “Wisconsins” should be “Wisconsin”
Lines 11267, 1133, and 1134: “Hawai’i University Press” should be “University of Hawai‘i Press”
Reviewer 2 Report
The article aims to present the interpretation of the tetralemma provided by Tiantai and Sanlun masters.
Sections 4 to 7 are highly interesting and to the point.
The introductory sections 1 to 3 are, however, problematic. Their style and level of clarity are quite different from the rest of the article and need a thorough revision.
They include a long discussion of the tetralemma in Indian philosophy.
In this discussion
1) the author basically refers to interpretations of modern scholars and does not consider a relevant recent publication (see my note on l. 312ff.) that has already discussed important point made by the author.
2) Indian philosophers are referred to by means of Chinese translations (especially in section no. 3), which poses a problem of method, namely whether the Chinese sources are reliable for reconstructing the ideas of Indian philosophers, or we should rather consider them as the Chinese interpretation of Indian texts that illustrated the thought of those philosophers, and investigate how and to what extent Chinese sources differ from Indian (or Tibetan) ones.
Therefore, e.g. in the case of Bhaviveka, Chinese sources should be presented more critically.
Especially with regard to Indian philosophers, the reader is not always clearly informed about the originality of the author’s analysis, namely, if and how it differs from previous scholarly work.
Several statements are common knowledge and are redundant in a research article (e.g., ll. 123-30, 359ff.)
Some statements are not precise and irrelevant to the argument made in the article > e.g., ll. 239f., 350f.
The abstract needs to be refined.
> the catuṣkoṭi is not a linguistic practice; translation too should be changed.
Remarks on specific points:
ll. 88-91 quickly discuss an important point. The actual discussion is in fn. 13, where, however, it is stated that Tiantai and Sanlun masters certainly follow a certainunderstanding, but the statement is not supported by quotations or interpretative reflections.
ll. 312ff. the argument is somehow similar to Priest’s interpretation, which has been criticized by Westerhoff (2020, p. 967). If the Chinese masters provide a different understanding which justifies the author’s statements, the author seems to have failed to explain this.
Translation of pratītyasamutpāda > conditioned origination, or conditioned arising (co-arising is an old-fashion translation that perhaps tried to reflect sam in the word samutpāda)
ll.144f. > tetralemma means “four alternative assumptions” (lemma means ‘assumption’ in Latin and Greek). It is an English word. It derives from Latin (where it was formed based on Greek words).
fn.6 > See also Jan Westerhoff, The Fifth Corner of Four: An Essay on Buddhist Metaphysics and the Catuṣkoṭi, by Graham Priest, Mind, Volume 129, Issue 515, July 2020, Pages 965–974, https://doi.org/10.1093/mind/fzz047
English language and style is to be revised > repetitions; redundant or unclear expressions, e.g., l. 304 correlative dependent opposites, l. 306 correlative dependent or interdependent opposites; l. 577f.; l.598f.; l. 995 “suspending four”; katophatic should be changed in cataphatic.
l. 129 > misspelling of Skt word
l. 285 > koṭis
Round 2
Reviewer 2 Report
I am happy to have contributed to this revised version of the present article. I’d be happy to see a more accurate way of reporting the reviewer’s opinion, if this is considered necessary. I think that a reviewer's comments should generate further reflection, in order to enhance clarity and quality of the discussion and arguments in the article.
Fn. 14: “One of the reviewers criticizes the use of “interdependence, mutuality, correlative opposites,” but this is what the Chinese “xiangdai相待” means.”
>>> I did not criticize “the use of “interdependence, mutuality, correlative opposites.”
Rather, I wrote that there are “redundant or unclear expressions, e.g., l. 304 correlative dependent opposites, l. 306 correlative dependent or interdependent opposites”.
It seems to me that “mutually dependent or correlative opposites” are better ways of translating the Chinese expression.
By the way, to translate the meaning of words and sentences means to find ways to adequately render those words and sentences in a source language into a target language. There is no “given” meaning in a language. The Chinese x means what the translators made it to mean over decades of centuries of translations.
Fn. 17: “Unlike the reviewer says, my concerns are different from Westerhoff’s critique, as they are directed against Priest’s thesis that paradoxical discourse in Madhyamaka expresses a certain metaphysical position.”
>>> I wrote with regard to ll. 312ff. (previous version; now ll. 344ff): “the argument is somehow similar to Priest’s interpretation, which has been criticized by Westerhoff (2020, p. 967).”
Revision of the English language in fn. 8.
Typos I noted:
- 245: ususally
end of fn. 8: Chinese, (Mitsuyoshi 1985)
- 17: necessiate
- 17: opposites, (see
- 20: philospher
- 20: interpretated
Author Response
I agree with the reviewer’s statement that the previous remarks helped improve the article, and, again, thank you very much for that contribution.
As to the points where I might have misrepresented the reviewer’s remarks, such as footnotes 14 and 17, I deleted the hint at the reviewer in those discussions.
I also tightened the English style in footnote 8 and corrected all the typos mentioned by the reviewer.
Again, thank you very much for the helpful and valuable comments.